# Reconstruct the Understanding of Grokking through Dynamical Systems

## Abstract

**Grokking**, or the **delayed generalization phenomenon**, describes the abrupt and rapid improvement in test accuracy that occurs after a model has been overfitted for a prolonged period. This phenomenon was first identified by Power in the context of operations on a prime number field. Over the past two years, a range of mathematical analyses has been conducted to investigate grokking, typically involving the use of the hidden progress measure which mean a function that can anticipate the occurrence of grokking. We believe that a comprehensive and rigorous mathematical modeling approach can invigorate the research on this task and provide a unified perspective for understanding previous research. This paper introduces a novel approach by modeling the task as a unique dynamical system. Using mathematical derivation within this framework, we propose a robust hidden progress measure that effectively captures the grokking phenomenon across all operations on prime number fields. This approach not only provides a more complete understanding but also offers deeper insights into the underlying architecture of the model. Based on this understanding, we also proposed a method to accelerate grokking without involving regularization or altering the model architecture.

## 1 Introduction

For a long time, it has been widely believed that model overfitting results from an excessive reliance on biases within the dataset, causing the model to lose its ability to generalize to new data. However, the observation of grokking (Power et al. (2022)) challenges this understanding. Grokking offers a new perspective on the training process, suggesting that improvements in training accuracy may only indicate a superficial "mastery" of the task by the model.

In recent years, numerous attempts have been made to explain grokking, typically following a pattern of hypothesis formulation - theorem construction - experimental validation. From a mathematical perspective, these efforts are phenomenological, lacking a deep exploration of the origins of the theoretical tools employed. To fully understand the grokking phenomenon, it may be necessary to undertake more structured work, specifically the development of a comprehensive mathematical model. Such a model would situate the problem within an established mathematical framework, thereby providing access to a variety of powerful analytical tools.

At first, Our work focused on placing the task where grokking happens, within the framework of dynamical systems. This allows us to utilize the tools of dynamical systems to examine the mathematical structure of the entire task. By studying the properties of the phase space of the dynamical system, we aim to further analyze the reasons behind the occurrence of grokking in these tasks. In Figure 1, we illustrate our understanding of the aforementioned concept, where we posit that work at the structural level should be prioritized.

A complete mathematical framework has allowed us to re-evaluate previous approaches to analyzing grokking. We acknowledge the existence of a hidden progress measure (Nanda et al. (2022)), which is a function embedded within the model's update process that can accurately represent grokking. We re-modeled the task using a dynamical systems approach, which allowed us to anchor the mathematical background of multiple hidden progress measures. Ultimately, based on our analysis of the specific dynamical system associated with this task, we have proposed a robust and elegant hidden progress measure. This measure can precisely track the occurrence of grokking and does not

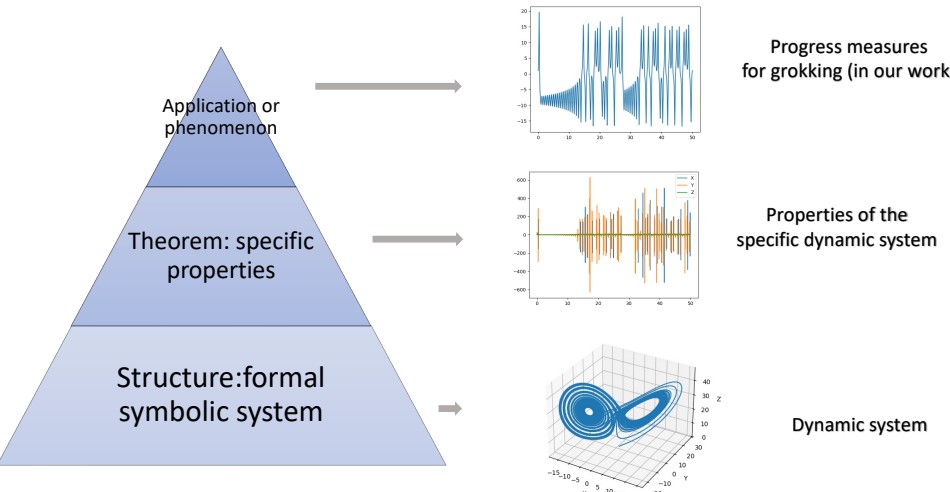

Figure 1: This figure is intended to illustrate the hierarchical levels of mathematical analysis. Structured work often involves introducing a problem into a well-established mathematical framework. Within this framework, general theorems can be combined with specific tasks to derive theorems that are applicable to those particular tasks. These task-specific theorems can then be directly employed to solve problems, representing the application-level aspect of the work.

depend on the selection of the prime number $p$ in the original task. We believe that the grokking phenomenon in this task is highly unique. Therefore, based on our theoretical framework, we attempted to accelerate grokking without using regularization techniques and achieved promising results.

The main contributions and innovations of our work can be summarized as follows:

- We re-modeled the prime field addition task from the perspective of dynamical systems and reconstructed the analysis of grokking.

- We proposed a function called Main embedding diff (MED) that can track changes in test loss, serving as a robust hidden progress measure. It is not limited to a specific prime number $p$, but can be applied to any task with a similar structure.

- Without using regularization techniques or altering the model architecture, we reduced the number of epochs required for grokking to half of the original.

## 2 RELATED WORK

### 2.1 GROKKING

Grokking was first proposed in Power et al. (2022), which happened on a model of a two-layers transformer decoder to solve several algorithmic tasks. Initial understanding of grokking focused on the size of the dataset and some studies proposed the concept of 'critical dataset size' like Zhu et al. (2024) and Huang et al. (2024). The essence of these methods is to explain mutation behavior through the linear variation of dataset size. Some studies have also recognized that grokking might be a common phenomenon in classification tasks, prompting researchers to approach the problem from a structural perspective like Liu et al. (2022) and Thilak et al. (2022) . However, the number of classification tasks in which the grokking phenomenon has been observed is limited, leading some to speculate that grokking is a result of the Transformer architecture, as discussed Wang et al. (2024). Some studies also focus on changing the architecture to investigate variations in the phenomenon like Park et al. (2024), Kunin et al. (2024) and Lee et al. (2024). And there have always been researchers who associate grokking with emergence, and notable work in this area includes: Mallinar et al. (2024), He et al. (2024), Zhao et al. (2024).

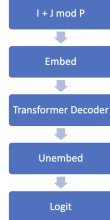 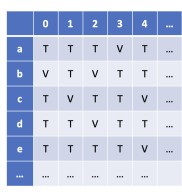

Figure 2: The left panel illustrates the architecture of our proposed model, while the right panel depicts the extended task structure. We conceptualize this as a natural language processing task. To achieve this, we employ $p$ symbols representing the range from 0 to $p-1$ (often select $k$ to $k+p-1$). Here, $T$ denotes elements within the training set, and $V$ denotes elements within the test set.

## 2.2 PROGRESS MEASURE

The concept of a progress measure was first introduced in Barak et al. (2022), essentially as a smooth function of continuous changes in activation values used to predict model behavior. One of perspectives is based on frequency and Fourier coefficients, inspired by circuit signal analysis, with representative papers including Nanda et al. (2022), Zhou et al. (2024) and Furuta et al. (2024). The second perspective is local complexity, derived from linear region analysis, with representative papers including Humayun et al. (2024). The third perspective is based on information theory and its representative paper is Clauw et al. (2024). Recently, some methods have aimed to provide a more unified perspective like Yunis et al. (2024) and Song et al. (2024).

## 2.3 NONLINEAR DYNAMICAL SYSTEM AND CHAOS

The primary mathematical framework we used is dynamical systems theory, and the mathematical aspects mainly reference the following sources: Birkhoff (1927), Malyshev (1993), Brin & Stuck (2002). Applying dynamical systems theory to optimization is a significant direction in modern control theory. An important outcome of this approach is gradient flow, with common results such as Gambarini et al. (2024), In recent years, there have also been efforts to integrate dynamical systems theory with large models, like Geshkovski et al. (2023) and Hernández & Zuazua (2024).

## 3 PRELIMINARIES

In Power et al. (2022), the task that exhibits the phenomenon of grokking involves using a transformer's decoder to learn addition mod $p$. The dataset is composed of $(i, j)$, $i, j \in \{0, 1, ..., p\}$, from which a subset is selected as the training set to test whether the model can generalize to the entire dataset.

In Figure 2, we present the model architecture and the modifications made to the task formulation based on our understanding. We consider this task to be understood as a classification task within the domain of natural language processing. Let $x_i$ represents the $i_{th}$ number's embedding vector, $p_i$ represents the $i_{th}$ position embedding vector. We can write the embedding process of $(i, j, p_+)$ as $(x_i + p_1, x_j + p_2, x_0 + p_3)$, and the last vector is used to simplify the complex embedding of $(+, mod p)$ . As the transformer block can be written as an affine transformation, and on the inverse-embedding layer, we use a matrix to map the activation values obtained from the previous transformation into an eigenvector,that means we can write this process as

$$y = Wx + b. \tag{1}$$

$$logit = softmax(W_U y). \tag{2}$$

$$\gamma = c - W_U W x. \tag{3}$$

## 4 THE DYNAMICAL SYSTEM MODEL

### 4.1 ESTABLISHMENT OF MODEL

To establish the dynamical system model, we first introduce some fundamental concepts. A dynamical system refers to a system that evolves over time under the influence of driving forces. It is one of the most commonly used mathematical models in control theory and optimization theory. There are many specific forms of representation for dynamical systems, such as difference equations, partial differential equations, and ordinary differential equations (ODEs). Here, we adopt the form of ordinary differential equations to describe the dynamical system.

**Definition 1** (Dynamical system). The following form of equations are what we refer to as a dynamical system

$$\frac{d\boldsymbol{x}}{dt} = f(\boldsymbol{x}, t; \mu),$$

$$\boldsymbol{x} \to g(\boldsymbol{x}; \mu),$$

with $\boldsymbol{x} \in \mathbb{U} \subset \mathbb{R}^n$, $t \in \mathbb{R}^1$, and $\mu \in \mathbb{V} \subset \mathbb{R}^p$, where $\mathbb{U}$ and $\mathbb{V}$ are open sets in $\mathbb{R}^n$ and $\mathbb{R}^p$.

The core of dynamical systems research lies in understanding the properties of phase space and the long-term behavior of trajectories.

**Definition 2** (Phase space and equilibrium solution). For a dynamical system $\frac{d\boldsymbol{x}}{dt} = f(\boldsymbol{x}, t; \mu)$, some interval $\mathbb{I} \subset \mathbb{R}^1$ into $\mathbb{R}^n$, which we represent as $\boldsymbol{x} : (\mathbb{I} \to \mathbb{R}^n)$ $t \to \boldsymbol{x}(t)$ with $\frac{d\boldsymbol{x}}{dt} = f(\boldsymbol{x}, t; \mu)$ satisfied. The map $\boldsymbol{x}$ called a trajectory and the space of the curve called the phase space of the dynamical system. The long-term behavior of dynamical systems is often closely related to their equilibrium points. Equilibrium points are these $\boldsymbol{x}$ make $f(\boldsymbol{x}, t; \mu) = 0$.

We can now introduce how general deep learning tasks can be modeled as dynamical systems. First, we need to understand the parameter update process as a temporal evolution, where each update represents a discrete time step. Consequently, all the model parameters are combined into the vector $\boldsymbol{x}$. The parameters driving the system's updates are derived from the training samples. For each sample, we can formulate the function $f(\boldsymbol{x}, t; \mu)$ determined by that sample according to the model's forward propagation method. A detailed description will be provided in Appendix A.

Returning to the phenomenon of grokking, as discussed in Section 3, a training sample is $(i, j, p_+)$ . Thus, the parameters updated by each sample point have overlaps. Therefore, if we assume that a time step corresponds to an epoch of parameter updates, it becomes challenging to formulate a complete function $f(\boldsymbol{x}, t; \mu)$, so we establish the dynamical system model corresponding to each individual sample for the task described in Section 3. Following the order of backward, the model can be formulated as follow:

$$\frac{d\boldsymbol{W_U}}{dt} = \sum_{i=0}^{p-1} \gamma_i \boldsymbol{e}^{(i)} (\boldsymbol{W}\boldsymbol{x})^T, \tag{4}$$

$$\frac{d\boldsymbol{W}}{dt} = \sum_{i=0}^{d} (\gamma \boldsymbol{W_U})_{i,:} \boldsymbol{e}^{(i)} \boldsymbol{x}^T, \tag{5}$$

$$\frac{d\boldsymbol{x}}{dt} = \gamma \boldsymbol{W_U} \boldsymbol{W}. \tag{6}$$

with $e_i$ represents the $i_{th}$ standard unit vector, $\boldsymbol{x}$ represents the vector obtained by concatenating the three embedding vectors corresponding to $(i, j, p_+)$. This model can be generalized to general classification tasks. Additionally, let us denote the reachable region $\Omega$ of the model's updating process as the state space of the dynamical system. $\Omega$ is a finite open set.

### 4.2 PROPERTIES FOR THE MODEL

One of the things we've been emphasizing is that each sample of a test set determines a corresponding dynamical system. In this task, due to the simplicity of the model architecture, these dynamical

systems have similar phase space structures (only one affine transformation from each other). Using this similarity, we can define a family of transformations that project a higher-dimensional phase space onto some lower-dimensional phase spaces ($\mathbb{R}^n \to \mathbb{R}^m$) to simplify our study of the properties of the system.

For the equation 4-6, we define a mapping family $\{F_{c,i,j,p}\}$ dependent on the entire sample set.

**Definition 3.** For the sample $(i, j, p_+) \to c$, we define a map $F_{s,i,j,p}, s = 1, 2, ..., p$ that it maps $\boldsymbol{x}, \boldsymbol{W}, \boldsymbol{W}_U$ in equation 4-6 into three scalars $x, w, u$ with equations

$$\frac{dx}{dt} = -(xwu - \delta_{s,c})wu, \tag{7}$$

$$\frac{dw}{dt} = -(xwu - \delta_{s,c})xu, \tag{8}$$

$$\frac{du}{dt} = -(xwu - \delta_{s,c})xw. \tag{9}$$

The $\delta_{s,c}$ mentioned above is a Kronecker symbol.

**Theorem 1.** *The map family defined above exists.*

The key idea of this theorem lies in quantifying the changes in task characteristics with model updates into a three-dimensional dynamical system. We prove this theorem using the Varison Lemma from functional analysis, thereby avoiding the cumbersome discussions in algebra. The detailed proof can be found in Appendix B. We consider the equilibrium points of the dynamical system defined by equation 4-6. Apart from the trivial equilibrium point, all equilibrium points satisfy $\gamma_i = 0$. Let $\mathbb{V}_c = \{\boldsymbol{x}|\boldsymbol{x}_c > \boldsymbol{x}_i, \forall i \neq c, 1 \leq i \leq p\}$. According to the softmax function employed in the model, essentially, all equilibrium points lie within the cone $\mathbb{V}_c$ as we have defined. We aim to thoroughly understand the properties of this dynamical system by mapping $\mathbb{V}_c \cup \Omega$ into a scalar quantity and then considering the process of updating this scalar. This approach allows us to use visualization techniques to comprehensively study the properties of the simplified system. Theorem 1 ensures the feasibility of this approach. Now, we will present the fundamental properties of the simplified dynamical system.

For the simplified dynamical system, we consider the properties of its trajectories within the unit cube $[0, 1]^3$.

**Theorem 2.** *The non-trivial attractors (equilibrium points) of the dynamical system defined by equation 7-9 are all stable.*

The proof of this theorem is quite straightforward, and we will place it in the appendix. This theorem means the dynamical system we study has simple structure of phase space, so the core problem is how to deal with the relationship between these mappings. To further understand this issue, we shall define the norm of the aforementioned family of mappings. The statement of this definition and the subsequent theorem require some background in the framework of functional analysis, which we will include in Appendix B.

**Definition 4.** $\|F_{s,i,j,p}\| = \sup\limits_{\boldsymbol{x} \in \mathbb{R}^n / \{\boldsymbol{0}\}} \{\frac{F_{c,i,j,p}(\boldsymbol{x})}{\|x\|_2}\}, (i, j, p_+) \to c.$

This definition will reflect the optimal transport path of the embedded vectors in the original embedding space. Based on this, we proceed to state our main result.

**Theorem 3.** *If the model have learnt all dynamical systems, then $\forall d_1 + d_2|p$, we have $\|F_{s,i,j,p} - F_{s,i+d_1,j+d_2,p}\| = 0$.*

This theorem provides the criteria for generalizing from the training set to the test set, we will see its proof in Appendix B.

**Assumption 1.** *For $(i, j, p+) \to c$ in test set V, consider the train set as T, the norm of $F_{s,i,j,p}$ satisfies*

$$\|F_{s,i,j,p}\| \propto \sum_{t_1,t_2 \in \mathbb{Z}/\{0\}, 0 \leq i+t_1 \leq p, 0 \leq j+t_2 \leq p} \frac{\alpha_{d_{model}}}{t_1^2 + t_2^2} \|F_{s,i+t_1,j+t_2,p}\|, \tag{10}$$

*with $\alpha_{d_{model}}$ is a function of $d_{model}$ and we guess it is approximately a linear function in specific situations.*

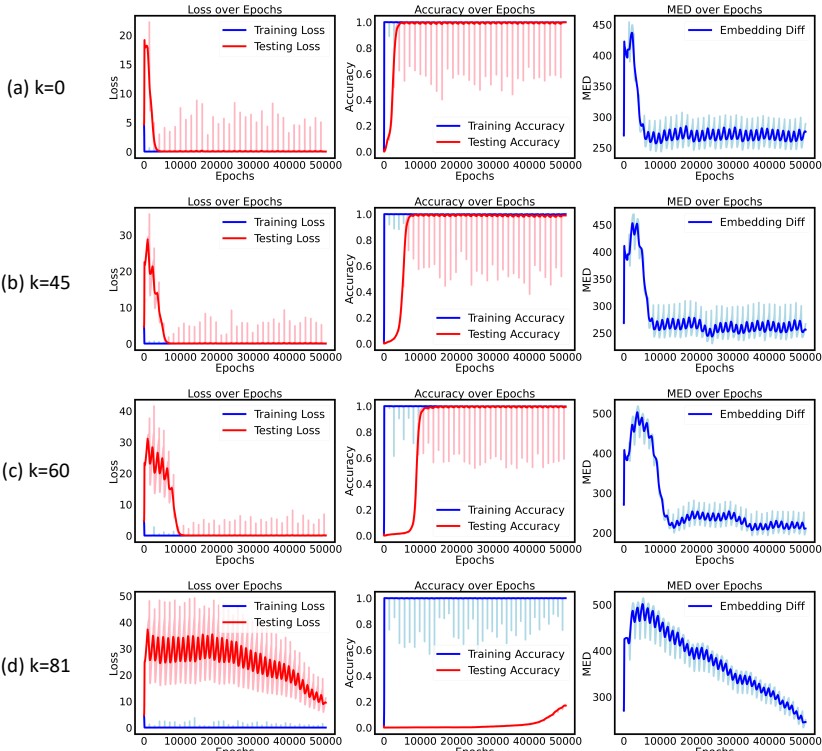

Figure 3: By reducing the range of overlap between $i$ and $j$, we observe the slowdown in the speed of grokking. By comparing with the lighter original images, we can directly see the ability of our designed MED function to track the test loss.

**Assumption 2.** *If the model have learnt all dynamical systems, then we have*

$$\|F_{s,i,j,p} - F_{s,i+t,p}\| = t\epsilon, \tag{11}$$

*with $\epsilon$ is a small positive number.*

### 4.3 MAIN EMBEDDING DIFF

In this section, we introduce our designed hidden progress measure that we call main embedding diff and its properties. Based on this, we provide an in-depth analysis of grokking, which will be validated in Section 5.

First, we provide a precise definition of the hidden progress measure for deep learning models. The focus is on formalizing the qualitative description of this concept presented in Barak et al. (2022).

**Definition 5** (Hidden progress measure)**.** Let the complete set of parameters of a deep learning model be denoted by $\mathbb{W}$, and the update step by $n$. Given a function $f : \mathbb{W} \times \mathbb{Z}_+ \to \mathbb{R}$, if there exists a mapping $\phi \circ f : \mathbb{W} \times \mathbb{Z}_+ \to \{0, 1\}$ such that $\phi \circ f$ takes the value of 1 when a specific phenomenon occurs and 0 otherwise, then we call $f$ a hidden progress measure of this specific phenomenon of the model.

We now introduce the methodology by which we derived our progress measure. We define an input region as $F_{c,i,j,p}^{-1}(\{1\})$ and interpret the task as a mapping from an input region to $\mathbb{V}_i$. The input region must satisfy the requirements outlined in Theorem 3.

**Theorem 4** (Main result)**.** *We divide the aforementioned task into two parts: the first part involves encoding using an embedding matrix, while the second part employs a transformer to decode and perform classification. The purpose of the latter part is to establish the region mapping described in Theorem 3, whereas the former part ensures that the input meets the requirements of the input region for the latter part.*

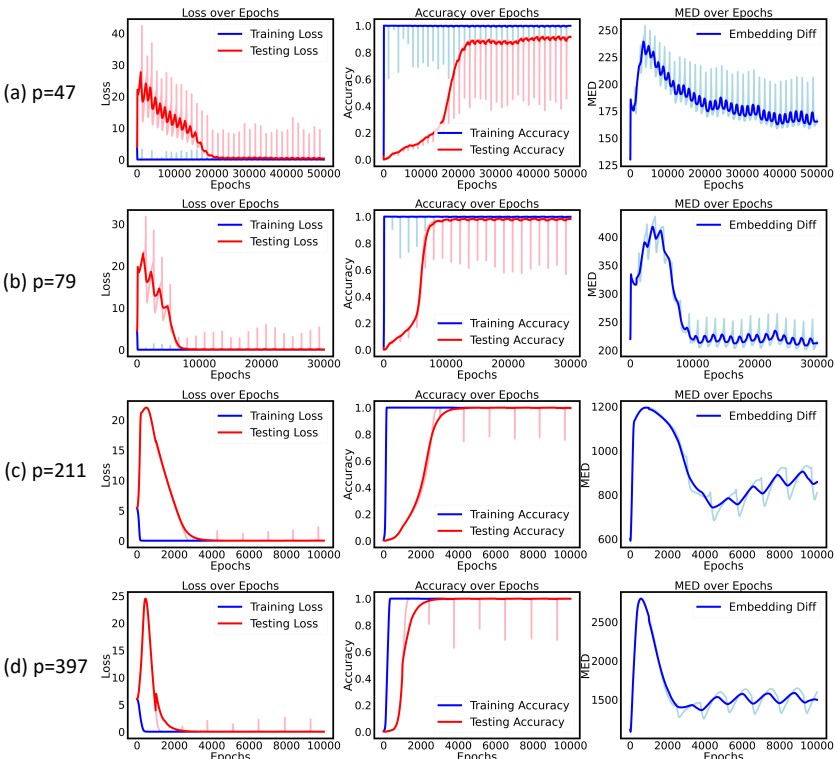

Figure 4: To more intuitively demonstrate the impact of changes in $p$, we reduced the number of epochs as $p$ increased. When $p = 47$, grokking still requires over 10,000 epochs, whereas when $p = 397$, it only needs 1,000 to 2,000 epochs.

In our task, the overall input distribution exhibits characteristics of a uniform distribution. Consequently, a single-layer transformer is the most stable, whereas a multi-layer transformer introduces complex structures in the input region, significantly increasing the proportion of training data required. This can lead to overfitting or even non-fitting issues, which we will verify in Experiment 4. We have verified that the input region distribution formed by the parameter updates of this model differs from the commonly understood concept of an embedding dictionary structure. For a detailed discussion, please refer to Appendix F.

Therefore, our progress measure is a quantity that can describe the characteristics of a uniform distribution, specifically aligning with Assumption 2.

**Definition 6** (Main embedding diff). In our task (as described in Figure 2), consider $1, 2, \ldots$ and $a, b, \ldots$, where these two p-tuples correspond to the sets of embedding vectors denoted as $\boldsymbol{x}_1^{(1)}, \ldots, \boldsymbol{x}_p^{(1)}$ and $\boldsymbol{x}_1^{(2)}, \ldots, \boldsymbol{x}_p^{(2)}$ respectively on the time step $n$, then the MAIN EMBEDDING DIFF of n defined as

$$\text{MED}(n) = \sum_{i=1}^{p-1} \left\| \boldsymbol{x}_i^{(1)} - \boldsymbol{x}_{i+1}^{(1)} \right\|_2 + \sum_{j=1}^{p-1} \left\| \boldsymbol{x}_j^{(2)} - \boldsymbol{x}_{j+1}^{(2)} \right\|_2. \tag{12}$$

We list the properties of the main embedding diff as follows.

**Pro 1.** *There exist a positive number $e$ and a positive integer $N$ such that, when $n > N$, $MED(n) < e$ and exhibits approximately periodic oscillations. This indicates that the model has reached the upper limit of its generalization capacity.*

We refer to property 1 as the restrictive nature of MED. The lower bound $e$ mentioned here often represents the upper bound of our task's generalization ability.

**Pro 2.** *Let the test loss be denoted by $TL(n)$, then $TL(n + 1) - TL(n)$ and $MED(n + 1) - MED(n)$ have the same sigh. This implies MED can track the variations in the test loss.*

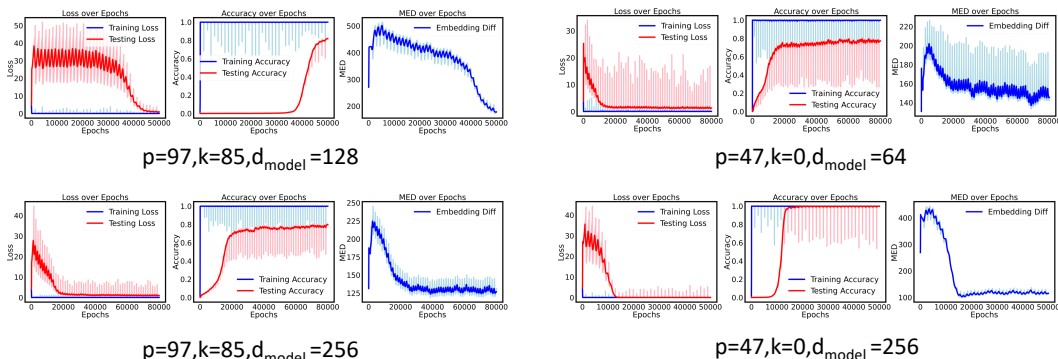

Figure 5: (Left) We selected $p = 97$ and $k = 85$ and we found that grokking occurred over 20,000 epochs earlier with $d_{model} = 256$ compared to $d_{model} = 128$. (Right) We selected $p = 47$ and $k = 0$ and we found that grokking occurred over 10,000 epochs earlier with $d_{model} = 256$ compared to $d_{model} = 64$.

The theory of dynamical systems can also accommodate previously established hidden progress measures like Fourier Decomposition in Nanda et al. (2022). We have included a detailed discussion of this aspect in Appendix C.

## 5 EXPERIMENTS

### 5.1 SETUP OF EXPERIMENTS AND OVERALL DESCRIPTION

Our experimental design is largely inherited from Power et al. (2022), and the basic setup remains the same. The only difference is Power used a transformer with 2 layers but we used a transformer with 1 layer for most of the experiments and the weight decay coefficient is set to 1.0.

All our experiments include three plots: a comparison of training/test loss, a comparison of training/test accuracy, and a plot showing the variation of the MED function. To better present the results, each dataset underwent filtering. The lighter color represents the plots before filtering, while the darker color indicates the plots after filtering.

All our experiments can demonstrate the overall tracking capability of our designed MED function for test loss. In all experiments, we observe the following phenomena:

- The original plot of the test loss and the original plot of the MED function exhibit an almost identical pattern of variation;

- When the MED function falls below its initial value, grokking happens;

- When the MED function begins to oscillate and no longer shows an overall downward trend, the increase in test accuracy also stops.

This also corresponds to the three properties we described earlier. This further explains the validity of Theorem 4. We also made some predictions based on the preceding theory and conducted ablation experiments. After completing the ablation study, we will provide a method to accelerate grokking.

### 5.2 MAIN EXPERIMENT

#### 5.2.1 EXPERIMENT: REDUCING THE OVERLAP IN THE DATA

In this experiment, we set $p = 97$ and gave different displacement on $i$ or $j$ when generating the dataset that dataset pairs turned into $(i, j + k)$. We selected $k = 0,45,60,81$ and we saw that the speed of grokking slows down as $k$ increases, Our main embedding diff can trace the val loss for all these situations whether the speed.

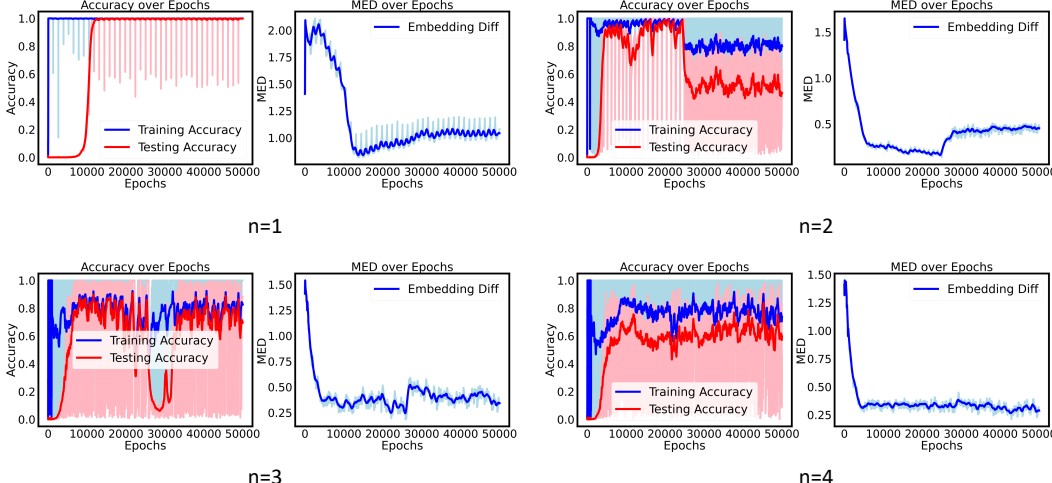

Figure 6: We use $n$ denote the number of transformer layers and replace MED with its mean value. The decrease in the final mean value of MED as the number of layers increases demonstrates the segmentation behavior of a multi-layer transformer on the input region. When $n = 1$, standard grokking was observed; at $n = 2$, overfitting occurred (Some studies refer to it as "ungrokking" (Varma et al. (2023)) while others term it "misgrokking" (Lyu et al. (2023)); and for $n = 3$ or $n = 4$, non-fitting was observed, The oscillatory behavior displayed with light coloring indicates that our fitting results are no longer of significant interest. However, when $n = 3$, an intermediate state between $n = 2$ and $n = 4$ emerges, which is evident in the U-shaped curve of our fitted results.

We designed this experiment based on Theorem 4, which indicates that an increase in $k$ means that the embedding layer needs to capture more distributional information, thereby increasing the learning time. The results can be saw in Figure 3.

### 5.2.2 EXPERIMENT: INCREASING THE SIZE OF THE DATASET

When grokking has not occurred, the norm of $F_{s,i,j,p}$ corresponding to points $(i, j, p_+)$ that are not in the training set tends to zero. Based on Assumption 1, we could conclude that

$$\|F_{s,i,j,p}\| \propto \sum_{t_1,t_2 \in \mathbb{Z}/\{0\}, 0 \leq i+t_1 \leq p, 0 \leq j+t_2 \leq p} \frac{\alpha_{d_{model}}}{t_1^2 + t_2^2} \|F_{s,i+t_1,j+t_2,p}\|$$

$$\propto \sum_{t_1,t_2 \in \mathbb{Z}/\{0\}, (i+t_1,j+t_2,p) \in T, 0 \leq i+t_1 \leq p, 0 \leq j+t_2 \leq p} \frac{\alpha_{d_{model}}}{t_1^2 + t_2^2} \|F_{s,i+t_1,j+t_2,p}\|.$$

The last expression has the upper and lower bounds as follows:

$$\frac{\alpha_{d_{model}} \beta p^2}{c^2} \leq \sum_{t_1,t_2 \in \mathbb{Z}/\{0\}, (i+t_1,j+t_2,p) \in T, 0 \leq i+t_1 \leq p, 0 \leq j+t_2 \leq p} \frac{\alpha_{d_{model}}}{t_1^2 + t_2^2} \|F_{s,i+t_1,j+t_2,p}\| \leq \frac{\alpha_{d_{model}} \beta p^2}{C^2},$$

with $\beta$ represents the proportion of the training set, and both $c$ and $C$ are constants.

This also implies that the size of the training set will determine the rate of grokking. Unlike previous work, we chose not to change the proportion of the training set but to vary the value of $p$ to demonstrate this result. Similarly, our designed MED function has demonstrated its capability across all values of $p$. The results could be saw in Figure 4.

### 5.2.3 EXPERIMENT: IMPACT OF MODEL WIDTH

According to our theory, another factor that influences the rate of grokking is the size of $d_{model}$. When grokking is certain to occur, changing $d_{model}$ can have an accelerating effect on grokking.

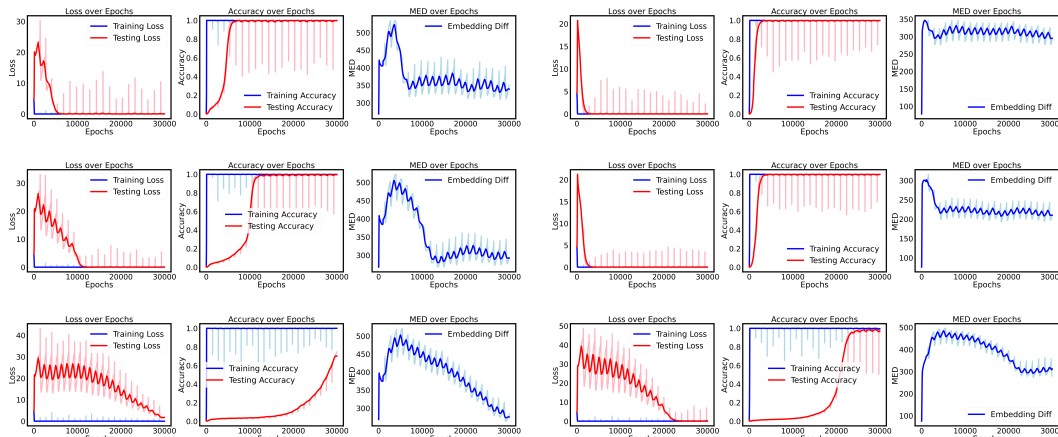

Figure 7: The experiments were conducted with $p = 97$, $k = 0$, and $d_{model} = 128$. The left side shows the results with random embeddings, while the right side displays the results using the improved embedding method. From top to bottom, the training set proportions are 0.3, 0.25, and 0.2, respectively. The number of epochs required for grokking to begin on the right side is only half of that on the left side.

In Figure 5, we present this result, with more detailed information provided in .

Additionally, we conducted extensive validation experiments, including various operations other than addition and the impact of positional encoding. These details are included in Appendix D.

### 5.2.4 EXPERIMENT: INCREASING THE NUMBER OF TRANSFORMER LAYERS.

We selected $p = 97$, $d = 97$, and $d_{\text{model}} = 512$, with a training set proportion of 0.3 to demonstrate the overfitting or even non-fitting phenomena that occur when increasing the number of transformer layers. The results could be saw in Figure 6.

### 5.3 ACCELERATION OF GROKKING

After thoroughly understanding the causes and monitoring methods of grokking, we will present an approach to accelerate grokking.

Since the occurrence of grokking implies thorough learning of the uniformity in the input space, our approach is to adjust the initial embedding values to facilitate this learning process. Specifically, we use a circulant matrix generated from a random vector to replace the embedding matrix, as expressed below:

$$\boldsymbol{W}_E = \text{Toeplitz}(v, v'), \tag{13}$$

with Toeplitz represents the operation of generating a Toeplitz matrix, and $v$ is a random vector. We have presented the experimental results in Figure 7.

## 6 CONCLUSION AND DISCUSSION

In this paper, we propose moving beyond a phenomenon-driven approach to theorem-based thinking and instead advocate for the use of structured mathematical tools. Based on this belief, we re-modeled the $i + j \mod p$ task using dynamical systems. Through theoretical analysis, we found that the cause of the grokking phenomenon is that, when the optimization task has a simple phase space structure, the speed at which the distribution of the task is learned is significantly slower than the speed at which the biases of the training set are learned. Additionally, we designed a powerful and concise hidden progress measure that can comprehensively track the test loss of a task under any condition. We believe that the intrinsic structure of the task is a crucial factor in the occurrence of grokking. Therefore, we achieved the acceleration of grokking without employing regularization techniques or modifying the model architecture.

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

## A   DYNAMICAL SYSTEM IN GENERAL TASKS

In this appendix, we present several more complex concepts in dynamic system modeling, employing purely formal notations to link the model with system optimization.

We consider a deep learning model with $n$ layers, where each layer corresponds to a manifold mapping $f_n(x; \omega_n)$ with $\omega_n$ represents the parameter set of the $n$-th layer. Here, the term "layer" refers to the component that induces the manifold mapping, and does not necessarily denote a specific layer. Let the loss function be denoted as $L$. Following the approach in this paper, we use measures to handle the dataset. We denote the embedding of the dataset, considering each token in the embedding layer as a high-dimensional vector in $\mathbb{R}^n$. We denote the empirical measure corresponding to these points in $\mathbb{R}^n$ as $\nu$.

**Definition 7.** The empirical measure $\mu$ in $\mathbb{R}^n$ for a dataset with $N$ data points is defined as:

$$\mu = \frac{1}{N} \sum_{i=1}^{N} \delta_{\boldsymbol{x}_i},$$

where $\boldsymbol{x}_i \in \mathbb{R}^n$ denotes the high-dimensional vectors corresponding to each data point, and $\delta \boldsymbol{x}_i$ represents the Dirac measure centered at $\boldsymbol{x}_i$.

We denote the functional relationships involved in the backward process as $\partial L$ and $\partial f_n$, the model's training process is equivalent to the following equation:

$$\frac{d\nu}{dt} = \prod_{i=1}^{n} \partial f_i \partial L, \tag{14}$$

$$\frac{d\omega_k}{dt} = \prod_{i=k+1}^{n} \partial f_i \partial L. \tag{15}$$

Now, we still lack a method for the quantitative description of a task. We assume that the training (or pre-training) task of the model corresponds to a kernel function $K$ and a discriminative equation.

**Definition 8.** $K : \Omega \times \mathbb{R}^n \to \mathbb{R}$ is referred to as the kernel function corresponding to the training task if

$$\int_{\mathbb{R}^n} K(x, \prod_{i=1}^{n} f_i \circ d\nu) = 1, \tag{16}$$

with $x \in \Omega$ means the model training has succeeded.

The issues corresponding to the phenomena of delayed generalization and emergence can be formulated as follows:

**Definition 9** (Delayed generalization). For the family of empirical measures $\{\nu_i\}_{i \in \{1,...,N\}}$, delayed generalization means there exist a positive integer $n < N$ such that:

$$\int_{\mathbb{R}^n} K(x, \prod_{i=1}^{n} f_i \circ d\nu_n) = 1 \Leftrightarrow \int_{\mathbb{R}^n} K(x, \prod_{i=1}^{n} f_i \circ d\nu_N) = 1. \tag{17}$$

**Definition 10** (Emergence). For the family of kernel functions $\{K_i\}_{i \in \{1,...,N\}}$, emergence means there exist a positive integer $n < N$ such that:

$$\int_{\mathbb{R}^n} K_n(x, \prod_{i=1}^{n} f_i \circ d\nu) = 1 \Leftrightarrow \int_{\mathbb{R}^n} K_i(x, \prod_{i=1}^{n} f_i \circ d\nu) = 1, \forall i \in \{1...N\}. \tag{18}$$

Our work is guided by the framework outlined above. In our approach, we use dimensional reduction to transform the integral of the kernel function equaling 1 into $uwx = 1$. This is because we were unable to find a method that fully characterizes the kernel function, which represents the greatest challenge encountered in this modeling process.

# B   PROOF OF THEOREM AND VISUALIZATION

## B.1   PROOF OF THEOREM 1 AND THEOREM 3

Theorem 1 is an existence theorem, which is used to ensure the existence of the framework for our method.

We employ Urysohn's Lemma from topology to establish the existence result here. Urysohn's Lemma is a classical tool in topology, often used to demonstrate the construction of specific types of continuous functions within certain topological spaces.

**Lemma 1** (Urysohn's lemma). *Let $\mathbb{X}$ be a normal topological space, and let $\mathbb{A}$ and $\mathbb{B}$ be two disjoint closed subsets of $\mathbb{X}$. Then, there exists a continuous function $f : \mathbb{X} \to [0, 1]$ such that $f(\mathbb{A}) = \{0\}$ and $f(\mathbb{B}) = \{1\}$*

Remember that $\mathbb{V}_c = \{\boldsymbol{x} | \boldsymbol{x}_c > \boldsymbol{x}_i, \forall i \neq c, 1 \leq i \leq p\}$, $\boldsymbol{W}_u \boldsymbol{W} \boldsymbol{x}$ defined by equation 1-3 is a vector in $\mathbb{R}^p$. After the model training is completed, we consider the convex hull of all vectors that lie within the cone $\mathbb{V}_c$, denoted as $\mathbb{B}_c$. And the complement of $\mathbb{V}_c$ in $\bar{\Omega}$ denoted as $\mathbb{A}_c$.

Since $\mathbb{B}_c \subset\subset \mathbb{V}_c$, $\mathbb{A}_c$ and $\mathbb{B}_c$ are two separable closed sets. Moreover, since every Euclidean space is a normal space, we can invoke Urysohn's lemma. Thus, there exists a continuous function $f : \mathbb{R}^p \to [0, 1]$ such that $f(\mathbb{A}_c) = \{0\}, f(\mathbb{B}_c) = \{1\}$. For a vector $\boldsymbol{W}_u \boldsymbol{W} \boldsymbol{x}$ in $\mathbb{R}^p$, we denote $f(\boldsymbol{W}_u \boldsymbol{W} \boldsymbol{x}) = uwx$.

Now, we will employ a similar approach to derive the complete mapping that we require. Consider the vector space $\mathbb{R}^n$ and closed sets $\mathbb{A}'_c, \mathbb{B}'_c$ such that $\boldsymbol{W}_u \mathbb{A}'_c = \mathbb{A}_c$, $\boldsymbol{W}_u \mathbb{B}'_c = \mathbb{B}_c$. Using Urysohn's lemma like above we obtain there exists a continuous function $f' : \mathbb{R}^n \to [0, 1]$ such that $f'(\mathbb{A}'_c) = \{0\}, f'(\mathbb{B}'_c) = \{1\}$. For a vector $\boldsymbol{W} \boldsymbol{x}$ in $\mathbb{R}^n$, we denote $f'(\boldsymbol{W}_u \boldsymbol{W} \boldsymbol{x}) = wx$. Similarly, we can obtain that the function $f'' : \mathbb{R}^n \to [0, 1]$ and denote $f''(\boldsymbol{x}) = x$.

According to our definition, when the model makes a correct prediction, $x = w = u = 1$; when the model makes an incorrect prediction, $x = 0$. This satisfies the equation 7-9 when $s = c$. When $s \neq c$, we can use a similar approach by extracting a closed set from the high-dimensional cone $\mathbb{V}_s$ and applying the Urysohn lemma to it along with the remaining set. It should be noted that this time the complement set is mapped to 1, while the subset is mapped to 0.

And the proof of Theorem 3 is included in the proof of Theorem 1 where we use the same subset of $\mathbb{V}_c$ to conduct $F_{s,i,j,p}$ and $F_{s,i+d_1,j+d_2,p}$. To derive our proposed progress measure, we further introduce two assumptions which we believe are correct based on the information compression capability of manifold learning.

### B.2 PROOF OF THEOREM 2 AND VISUALIZATION OF ITS PHASE SPACE

Theorem 2 aims to demonstrate that the compressed dynamical system within the unit cube exhibits simple trajectory properties, thereby indicating that the complex, high-dimensional nature of the dynamical system is not a phenomenon that requires specific consideration for this task.

We take the following system of equations as an example, with similar reasoning applying to other cases.

$$\frac{dx}{dt} = (1 - xwu)wu, \tag{19}$$

$$\frac{dw}{dt} = (1 - xwu)xu, \tag{20}$$

$$\frac{du}{dt} = (1 - xwu)xw. \tag{21}$$

We first provide the precise definition and the criteria for determining the stability of equilibrium points in dynamical systems.

**Definition 11.** Consider an equilibrium point $\boldsymbol{x}^*$ of a dynamical system.

- $\boldsymbol{x}^*$ is said to be stable if, for any $\epsilon > 0$, there exists a $\delta > 0$ such that if $\|\boldsymbol{x}(0) - \boldsymbol{x}^*\| < \delta$, then $\|\boldsymbol{x}(t) - \boldsymbol{x}^*\| < \epsilon$ for all $t \geq 0$.

- $\boldsymbol{x}^*$ is said to be asymptotically stable if it is stable and there exists a $\delta' > 0$ such that if $\|\boldsymbol{x}(0) - \boldsymbol{x}^*\| < \delta'$, then $\lim_{t \to \infty} \boldsymbol{x}(t) = \boldsymbol{x}^*$.

- $\boldsymbol{x}^*$ is said to be unstable if it is not stable; that is, there exists an $\epsilon > 0$ such that for any $\delta > 0$, there exists an initial condition $\boldsymbol{x}(0)$ with $\|\boldsymbol{x}(0) - \boldsymbol{x}^*\| < \delta$ but $\|\boldsymbol{x}(t) - \boldsymbol{x}^*\| \geq \epsilon$ for some $t \geq 0$.

To determine the stability category of an equilibrium point, the Lyapunov method is often used.

**Theorem 5** ( Lyapunov's stability theorem). *Consider a dynamical system described by:*

$$\frac{d\boldsymbol{x}}{dt} = f(\boldsymbol{x}), \quad \boldsymbol{x} \in \mathbb{R}^n,$$

*where $\boldsymbol{x}^*$ is an equilibrium point, i.e., $f(\boldsymbol{x}^*) = 0$. To determine the stability of the equilibrium point $x^*$, one constructs a Lyapunov function $V : \mathbb{R}^n \to \mathbb{R}$ that satisfies the following conditions:*

*1. $V(\boldsymbol{x}) > 0$ for all $\boldsymbol{x} \neq \boldsymbol{x}^*$, and $V(\boldsymbol{x}^*) = 0$ (positive definiteness).*

*2. The time derivative of $V(\boldsymbol{x})$ along the trajectories of the system, given by $\dot{V}(\boldsymbol{x}) = \nabla V \cdot f(\boldsymbol{x})$, satisfies $\dot{V}(\boldsymbol{x}) \leq 0$ (negative semi-definiteness).*

*Then:*

- *If there exists a Lyapunov function such that $\dot{V}(\boldsymbol{x}) < 0$ for all $\boldsymbol{x} \neq \boldsymbol{x}^*$, then the equilibrium point $\boldsymbol{x}^*$ is locally asymptotically stable.*

- *If $\dot{V}(\boldsymbol{x}) \leq 0$, then the equilibrium point is stable in the sense of Lyapunov, but not necessarily asymptotically stable.*

- *If no such Lyapunov function can be found, alternative methods must be used to analyze stability.*

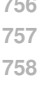

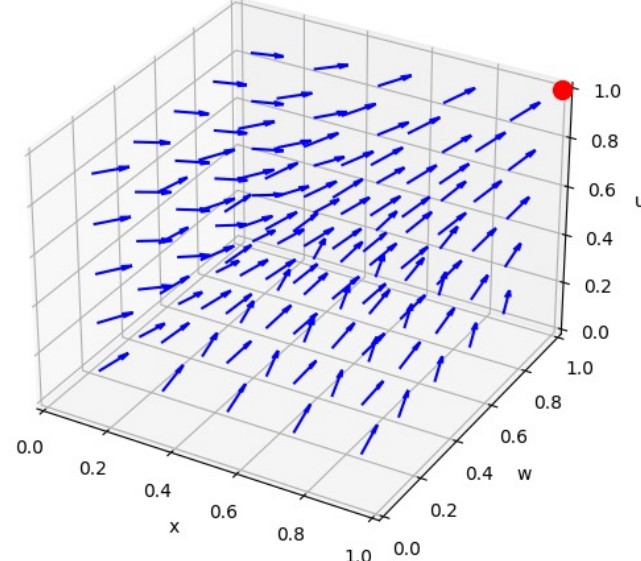

Visualization of the Dynamical System in the Unit Cube

Figure 8: This figure directly show the phenomenon of points within the unit cube moving towards the equilibrium point (1,1,1) which is highlighted.

For our equations we construct the function $V(x, w, u)$ as follows:

$$V(x, w, u) = (x - 1)^2 + (w - 1)^2 + (u - 1)^2. \tag{22}$$

It is obvious that $V > 0$ for all points in open unit cube and $V = 0$ when the system takes (1,1,1). Now we only need to verify that $\dot{V}$ is semi-positive definite within the unit cube.

$$\dot{V} = (3xwu - wu - xu - xw)(1 - xwu) \tag{23}$$

It is easy to verify that all of its leading principal minors are non-positive, hence it is non-positive definite. So (1,1,1) is a stable equilibrium point.

We can intuitively observe the simplicity of the structure of this dynamical system through visualization. We present the visualization results in Figure 8.

## C  DISCUSSION ON OTHER PROGRESS MEASURES IN DYNAMICAL SYSTEMS

In this appendix, we demonstrate how the two types of progress measures previously employed in the study of the grokking phenomenon are manifested within dynamical systems.

### C.1  FOURIER DECOMPOSITION

In the study of dynamical systems, the Fourier transform serves as a robust analytical tool for decomposing complex time-domain signals into their constituent frequency components. By converting a time-series signal into the frequency domain, the Fourier transform facilitates a deeper understanding of the periodicity and spectral characteristics inherent to the system under investigation. This method proves especially advantageous for identifying dominant frequencies and harmonics, which are crucial for elucidating the underlying mechanisms driving the system's behavior. In the context of learning algorithms, Fourier analysis can uncover latent patterns and progress indicators that

may not be evident in the time domain. For example, analyzing the Fourier gaps of the population gradient provides a novel framework for examining the convergence properties of the algorithm, allowing researchers to detect subtle improvements and oscillations that influence the overall dynamics of learning. Thus, the Fourier transform method enables a more nuanced and comprehensive understanding of both deterministic and stochastic processes in dynamical systems.

The following is some foundational knowledge required to address this problem.

### FOURIER TRANSFORM AND INVERSE FOURIER TRANSFORM

The Fourier transform is used to convert a time-domain (or spatial domain) signal into its frequency-domain representation. For a continuous function $f(t)$, its Fourier Transform is defined as:

$$F(\omega) = \int_{-\infty}^{\infty} f(t)e^{-i\omega t}\, dt,$$

with:

- $F(\omega)$ is the complex function in the frequency domain, representing the amplitude and phase at frequency $\omega$.

- $i = \sqrt{-1}$ is the imaginary unit.

- $\omega$ is the angular frequency (rad/s).

The inverse Fourier transform converts the frequency-domain signal back to the time-domain representation:

$$f(t) = \frac{1}{2\pi} \int_{-\infty}^{\infty} F(\omega)e^{i\omega t}\, d\omega.$$

### DISCRETE FOURIER TRANSFORM AND INVERSE DISCRETE FOURIER TRANSFORM

The discrete Fourier transform (DFT) is used to analyze the frequency components of a discrete signal. For a discrete signal of length $N$, $x[n]$, its DFT is defined as:

$$X[k] = \sum_{n=0}^{N-1} x[n]e^{-i\frac{2\pi}{N}kn}, \quad k = 0, 1, \ldots, N-1.$$

with:

- $X[k]$ is the complex value at discrete frequency $k$, representing amplitude and phase.

- $x[n]$ is the discrete signal in the time domain.

- $N$ is the number of sampling points.

The Inverse DFT is given by:

$$x[n] = \frac{1}{N} \sum_{k=0}^{N-1} X[k]e^{i\frac{2\pi}{N}kn}, \quad n = 0, 1, \ldots, N-1.$$

### FOURIER GAP

The Fourier gap describes the situation where certain frequency components are missing or filtered out in the frequency domain. In the presence of a Fourier Gap, the frequency-domain signal $G(\omega)$ can be expressed as:

$$G(\omega) = F(\omega) \cdot W(\omega),$$

with:

- $F(\omega)$ is the Fourier transform of the original signal.

- $W(\omega)$ is a window function that determines which frequency components are retained ($W(\omega) = 1$) or filtered out ($W(\omega) = 0$).

A Fourier gap may cause information loss or spectral distortion during signal reconstruction.

DECAY OF FOURIER COEFFICIENTS

The rate of decay of Fourier coefficients reflects the distribution of the signal in the frequency domain and is closely related to its smoothness in the time domain.

**Theorem 6** (Decay of Fourier coefficients for periodic signals). *If $f(t)$ is a periodic function with period $2\pi$ and $k$-times continuously differentiable, then its Fourier series coefficients $c_n = a_n + ib_n$ satisfy:*

$$|c_n| \leq \frac{C}{|n|^k},$$

*where $C > 0$ is a constant.*

This implies that the smoother the function (higher order $k$), the faster the decay of the Fourier coefficients $|c_n|$.

**Theorem 7** (Decay of Fourier transform for non-periodic signals). *If $f(t)$ is an absolutely integrable function with a $k$-th order continuous derivative $f^{(k)}(t)$, and its derivative is also absolutely integrable, then its Fourier Transform $F(\omega)$ satisfies:*

$$|F(\omega)| \leq \frac{C}{|\omega|^k},$$

*where $C > 0$ depends on $f(t)$ and its derivatives.*

This theorem indicates that if a signal is sparse in some basis (such as the Fourier basis), the number of measurements required for reconstruction can be much smaller than traditionally required.

A METHOD FOR RECONSTRUCTING FOURIER COEFFICIENTS AS A MEASURE OF PROGRESS.

In dynamical systems, a signal generally refers to a state variable that changes over time. In the context of deep learning models modeled as dynamical systems, it refers to a scalar or vector-valued function with model weights as its independent variables.

$x$, $W$, $W_u$ are all treated as signals in Nanda et al. (2022), they found that the Fourier coefficients in this context exhibit significant sparsity, According to Theorems 6 and 7, the emergence of sparsity implies that the signal exhibits increasing periodicity or quasi-periodicity. Our method has mentioned that, upon the completion of generalization, the embedding exhibits norm-based periodicity, which coincides with the results observed in the Fourier decomposition.

COMPRESSED SENSING THEORY

We believe that the observation of Fourier sparsity is likely to have broad applications beyond this specific task; it may be a common phenomenon within Transformer architectures. This is because the Transformer itself is a powerful tool for information compression and reconstruction. Therefore, we have included the well-known perceptual reconstruction theorem below in the hope that it will provide valuable insights.

**Theorem 8** (Compressed sensing reconstruction). *Let $x \in \mathbb{R}^N$ be a $K$-sparse signal (i.e., with only $K$ non-zero Fourier coefficients). Using $M = O(K \log(N/K))$ linear measurements $y = Ax$ (where $A \in \mathbb{R}^{M \times N}$ is a random measurement matrix), the signal $x$ can be exactly reconstructed using $\ell_1$-minimization algorithms.*

## C.2 LOCAL COMPLEXITY

Unlike Fourier decomposition, which aligns directly with the classical approach to studying dynamical systems, the concept of local complexity corresponds more closely to the method known as Poincaré sections in dynamical system analysis.

The Poincaré section method is a fundamental technique in the study of dynamical systems, offering a way to reduce the dimensional complexity of continuous systems by examining their intersections with a lower-dimensional subspace. This method involves selecting a hyperplane, referred to as the Poincaré section, which the system's trajectories intersect transversely. By analyzing the sequence of intersection points, one can study discrete dynamics and capture essential features of the system's behavior, such as periodic orbits, quasi-periodic motions, and chaotic trajectories. The Poincaré section thus serves as a powerful tool for visualizing and understanding the qualitative nature of dynamical systems, providing insights into stability, bifurcations, and the global structure of phase space. In the context of learning algorithms and complex systems, applying Poincaré sections helps identify invariant sets and analyze long-term behavior, thereby enhancing our capacity to predict and control system dynamics.

**Definition 12.** Let $\Phi^t : \mathbb{M} \to \mathbb{M}$ denote a smooth dynamical system defined by the flow $\Phi^t(\boldsymbol{x})$, where $\boldsymbol{x} \in \mathbb{M}$ and $\mathbb{M}$ is an $n$-dimensional differentiable manifold (phase space). A Poincaré section $\Sigma \subset \mathbb{M}$ is a codimension-one submanifold (i.e., of dimension $n - 1$) such that:

1. $\Sigma$ is transverse to the flow $\Phi^t(\boldsymbol{x})$, meaning that at every point $\boldsymbol{x} \in \Sigma$, the flow vector $\dfrac{d}{dt}\Phi^t(\boldsymbol{x})$ is not tangent to $\Sigma$.

2. Every trajectory of the flow $\Phi^t(\boldsymbol{x})$ that starts near $\Sigma$ will intersect $\Sigma$ again after some time.

**Definition 13.** The Poincaré map (or first return map) $P : \Sigma \to \Sigma$ is a discrete dynamical system defined by the intersections of the trajectories of the original flow with the Poincaré section $\Sigma$. For a point $\boldsymbol{x} \in \Sigma$, the Poincaré map $P(\boldsymbol{x})$ is the point of the next intersection of the trajectory through $x$ with $\Sigma$. Formally,

$$P(\boldsymbol{x}) = \Phi^{T(\boldsymbol{x})}(\boldsymbol{x}),$$

where $T(\boldsymbol{x}) > 0$ is the smallest positive time such that $\Phi^{T(\boldsymbol{x})}(\boldsymbol{x}) \in \Sigma$.

**Theorem 9.** *Let $\boldsymbol{x}^* \in \Sigma$ be a fixed point of the Poincaré map $P : \Sigma \to \Sigma$, i.e., $P(\boldsymbol{x}^*) = \boldsymbol{x}^*$. Then, $\boldsymbol{x}^*$ corresponds to a periodic orbit of the original flow $\Phi^t$. The stability of the periodic orbit is determined by the eigenvalues of the derivative $DP(\boldsymbol{x}^*)$ of the Poincaré map at $\boldsymbol{x}^*$.*

*If all the eigenvalues of $DP(\boldsymbol{x}^*)$ have magnitudes less than one, the periodic orbit is stable; if any eigenvalue has a magnitude greater than one, the periodic orbit is unstable.*

In light of the local complexity defined in Humayun et al. (2024), this concept treats the hierarchy of deep networks as a nested chain of affine transformations. Let us consider an affine transformation from one layer to the next, given by $\boldsymbol{y} = \boldsymbol{W}\boldsymbol{x} + \boldsymbol{b}$. Using $\boldsymbol{w}_i$ to represent the i-th row vector of $\boldsymbol{W}$, we can define a family of hyperplanes $\{\boldsymbol{w}_i x + b_i = 0\}$. The local complexity is then expressed by the number of hyperplanes intersecting the convex hull formed by the samples.

The Poincaré section indicates that an increase in local complexity corresponds to an increase in the density of trajectories on the hyperplane. When the trajectory density approaches a stable value, it often signifies that the dynamical system has converged to a stable solution, representing the completion of generalization.

# D SUPPLEMENTARY EXPERIMENTS

In this appendix, we conduct a series of supplementary experiments to discuss additional boundary conditions.

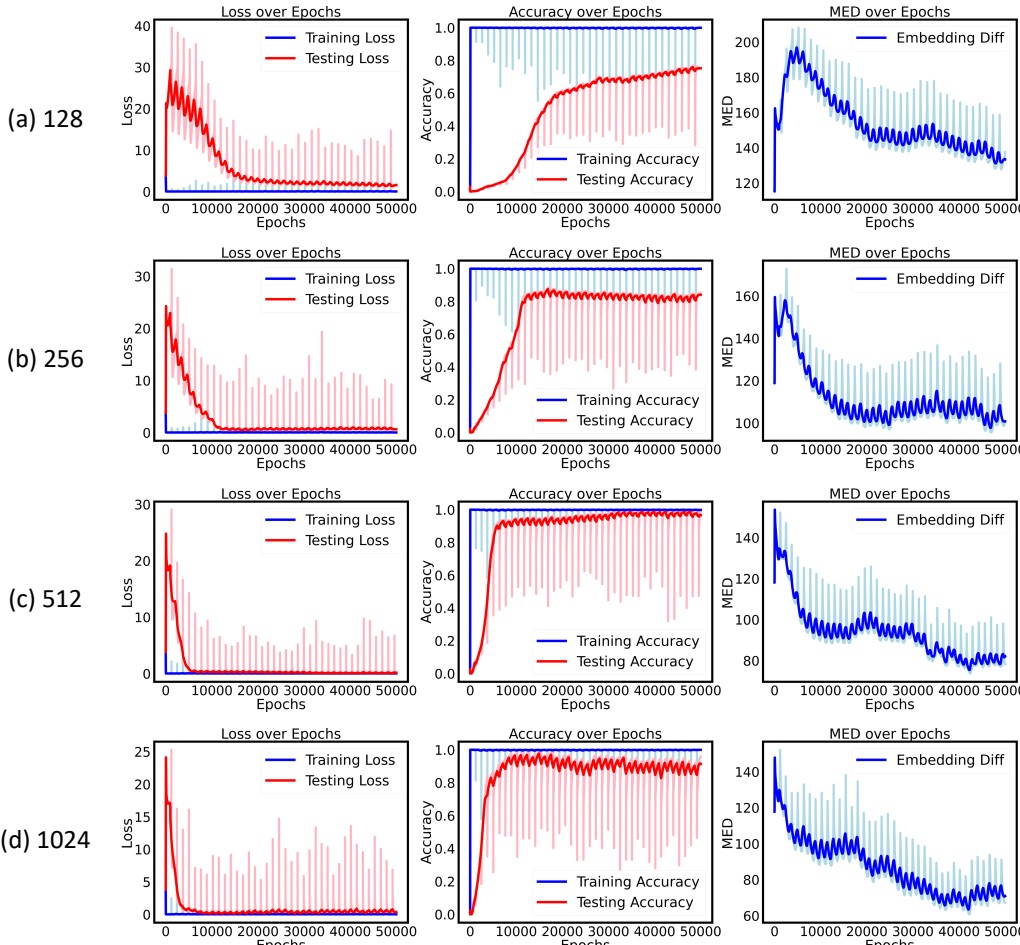

Figure 9: This figure aims to compare the impact of different embedding dimensions on the speed and upper limit of generalization. We selected the same number of epochs for all cases. We found that although increasing the embedding dimension from 512 to 1024 resulted in a decrease in the lower bound of the MED from around 80 to around 60, there was no significant improvement in generalization ability.

## D.1 THE UPPER LIMIT OF GENERATION: SIZE OF DATASET OR WIDTH OF MODEL

Our first supplementary experiment is designed to investigate the following question: when the dataset size is very small, is it possible that increasing the embedding dimension of the model could trigger the phenomenon of grokking?

We first selected the case where $p = 43$ and $k = 0$. Unlike the experiments conducted earlier (Figure 5), we can clearly observe that as the embedding dimension of the model increases, the final upper bound of our test accuracy also increases. When the embedding dimension is set to 128, the test accuracy oscillates around 0.7. However, when the embedding dimension is increased to 512, the test accuracy ultimately rises to 0.99. We present the results in Figure 9.

However, the results mentioned above are not at the boundary. To observe the boundary results, we continuously increased the value of $k$ until the grokking phenomenon no longer appeared at lower embedding dimensions. Then, we further increased the embedding dimension. The results were remarkable: increasing the embedding dimension indeed led to the appearance of the grokking phenomenon. This further supports the claim of our Theorem 4. Although Property 2 of the MED (Minimum Embedding Dimension) has already indicated that when it is reduced to a certain level, the model's generalization ability has reached its limit, we still increased the number of epochs to

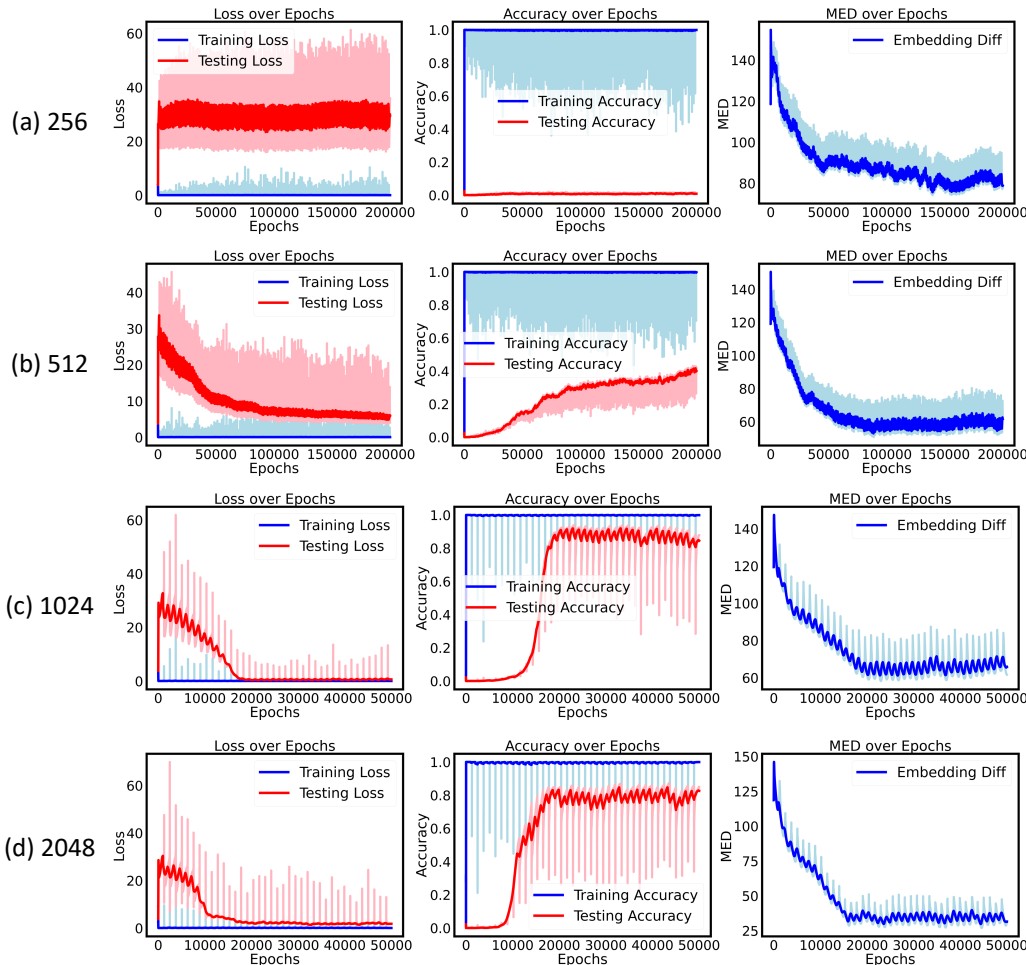

Figure 10: We discarded the case with an embedding dimension of 128 because it was identical to the plot for the 256-dimensional case. Note that we extended the number of epochs for the 256 and 512 dimensions to 200,000 to demonstrate that they had already reached the upper limit of their generalization abilities. When the embedding dimension was increased from 1024 to 2048, we observed a similar outcome to the previous experiment, with no noticeable change.

verify that grokking does not occur, rather than just not having occurred yet. We present these results in Figure 10.

We further discuss the boundary case where $p = 43$ and $k = 43$. Our experiments suggest that, assuming there is a metric for measuring the generalization ability of a model, it is inevitably influenced by parameters that are closely related to the prior distribution of the data. Similarly, we hypothesize that the further reduction of the MED after the generalization performance reaches its upper limit is likely due to capturing redundant details.

### D.2 OTHER OPERATIONS OVER PRIME FIELDS

We also verified the performance of the MED function on several other operations. We selected five cases to represent this: $x - y$, $xy$, $x^2 + y^2$, $x^2 + xy + y^2$, $x^3 + y^3$, and $x^3 + xy + y^3$.

Through the experiments, we found that our MED function is still exceptionally capable of tracking test loss. However, the grokking phenomenon is not always consistent. While subtraction still exhibits the grokking phenomenon, the generalization of second-order operations is highly dependent on the dataset size. This is essentially due to the inherent difficulty in generalizing nonlinear relationships without prior assumptions. For detailed experiments, please refer to Figures 11 to 18.

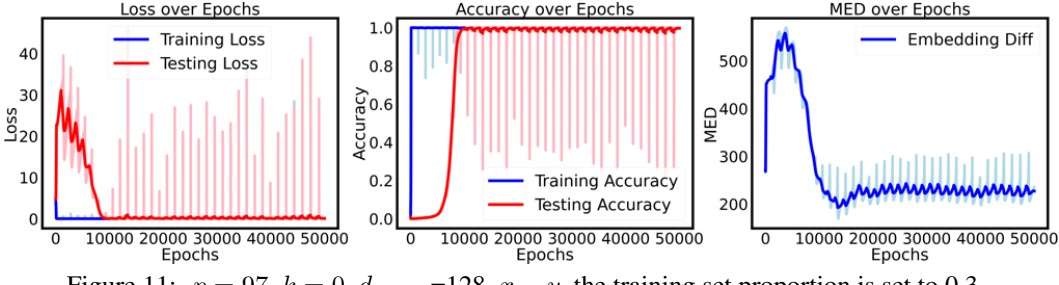

Figure 11: $p = 97$, $k = 0$, $d_{model}$=128, $x - y$, the training set proportion is set to 0.3.

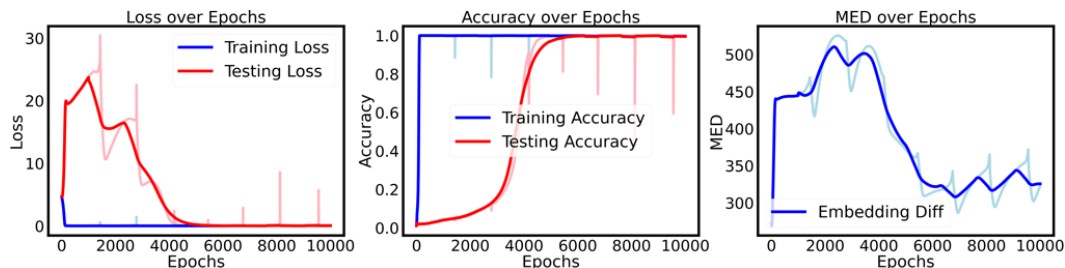

Figure 12: $p = 97$, $k = 0$, $d_{model}$=128, $xy$, the training set proportion is set to 0.3.

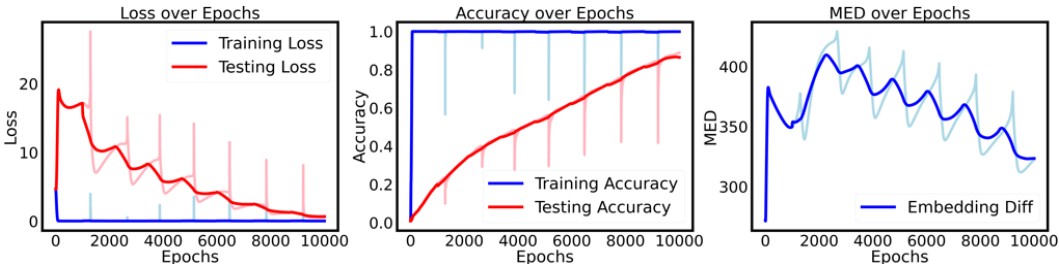

Figure 13: $p = 97$, $k = 0$, $d_{model}$=256, $x^2 + y^2$, the training set proportion is set to 0.2.

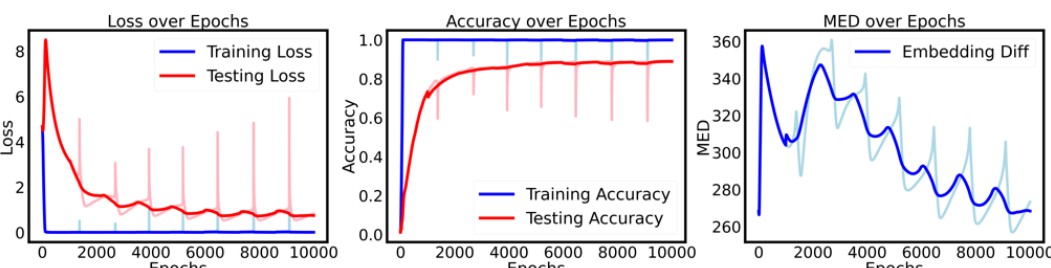

Figure 14: $p = 97$, $k = 0$, $d_{model}$=128, $x^3 + y^3$, the training set proportion is set to 0.2.

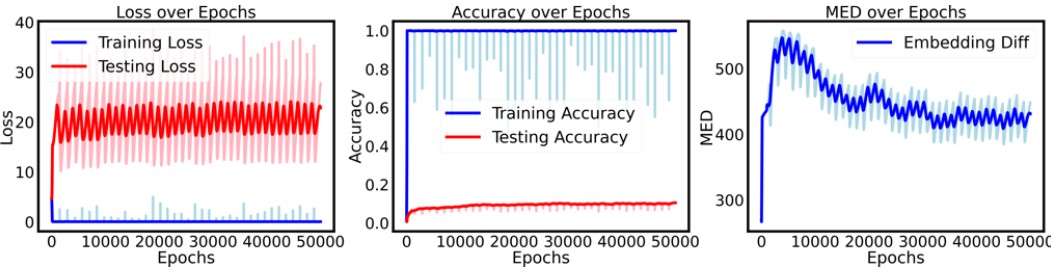

Figure 15: $p = 97$, $k = 0$, $d_{model}$=128, $x^2 + xy + y^2$, the training set proportion is set to 0.3.

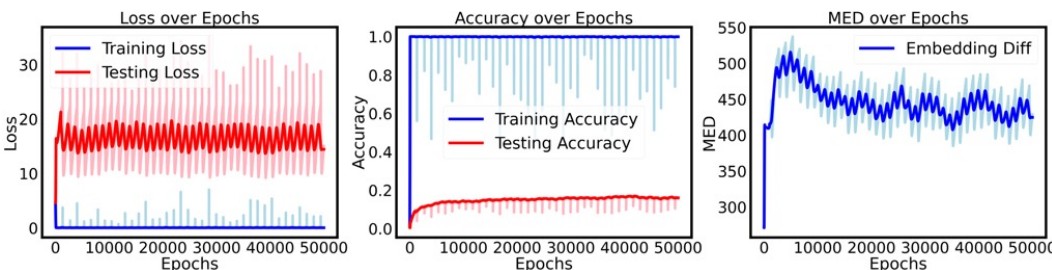

Figure 16: $p = 97$, $k = 0$, $d_{model}$=256, $x^2 + xy + y^2$, the training set proportion is set to 0.3.

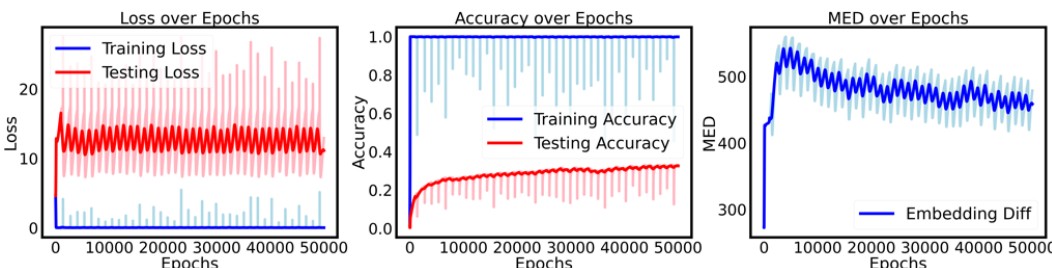

Figure 17: $p = 97$, $k = 0$, $d_{model}$=256, $x^2 + xy + y^2$, the training set proportion is set to 0.4.

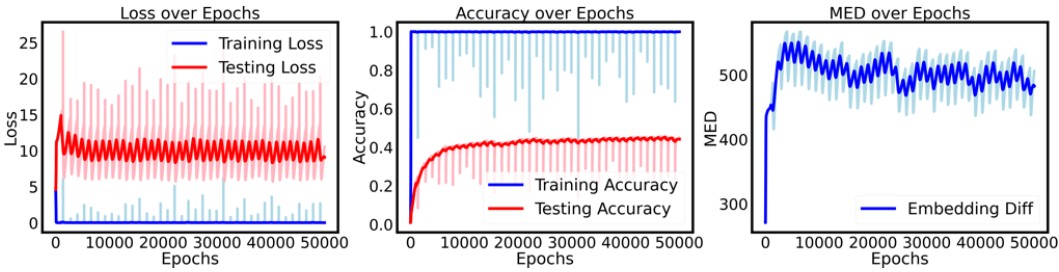

Figure 18: $p = 97$, $k = 0$, $d_{model}$=256, $x^3 + xy + y^3$, the training set proportion is set to 0.5.

## E  SOME ANOMALOUS SITUATIONS IN EXPERIMENTS

In the final appendix, we report the abnormal phenomena observed when the model fails to fully generalize. In our analysis, these phenomena share the same origin as the occasional confusion exhibited by large models.

### E.1  CHALLENGES POSED BY POSITIONAL ENCODING

We found that the use of positional encoding can effectively lower the lower bound of MED, which is quite a natural outcome. This is because it essentially prescribes a certain offset value for each position in advance, thereby compensating for some of the MED's functionality. Additionally, due to the commutative nature of the computational structure, the use of positional encoding should not affect grokking. However, experiments show that using positional encoding often results in a delay in grokking. We demonstrate this general phenomenon in Figure 19 by selecting the case where $p = 97$, $k = 97$, $d_{model} = 128$ and the training set proportion is set to 0.3.

This phenomenon is undoubtedly difficult to understand. We believe it is due to instability caused by the small dataset size. Therefore, we increased the embedding dimension of the model, and when $d_{model} = 256$, an even more peculiar phenomenon emerged, which we present in Figure 20.

The solution to this instability is quite simple: by increasing the embedding dimension to 512, this instability disappears. We present the results in Figure 21.

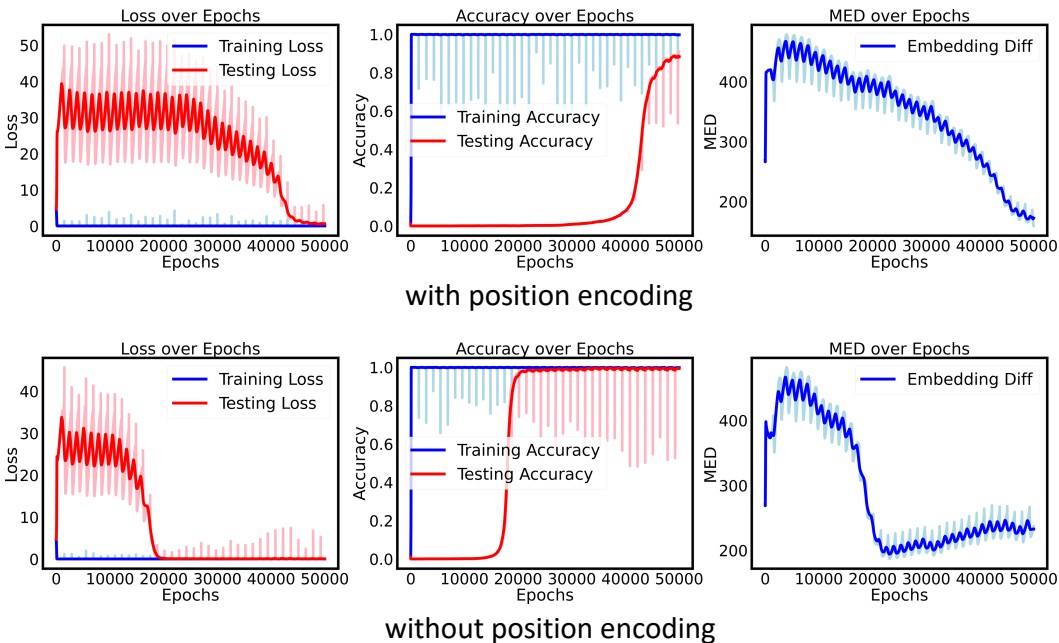

Figure 19: The upper figure represents the scenario where positional encoding is used, while the lower figure shows the scenario without positional encoding. The difference in rates between these two cases is remarkably significant.

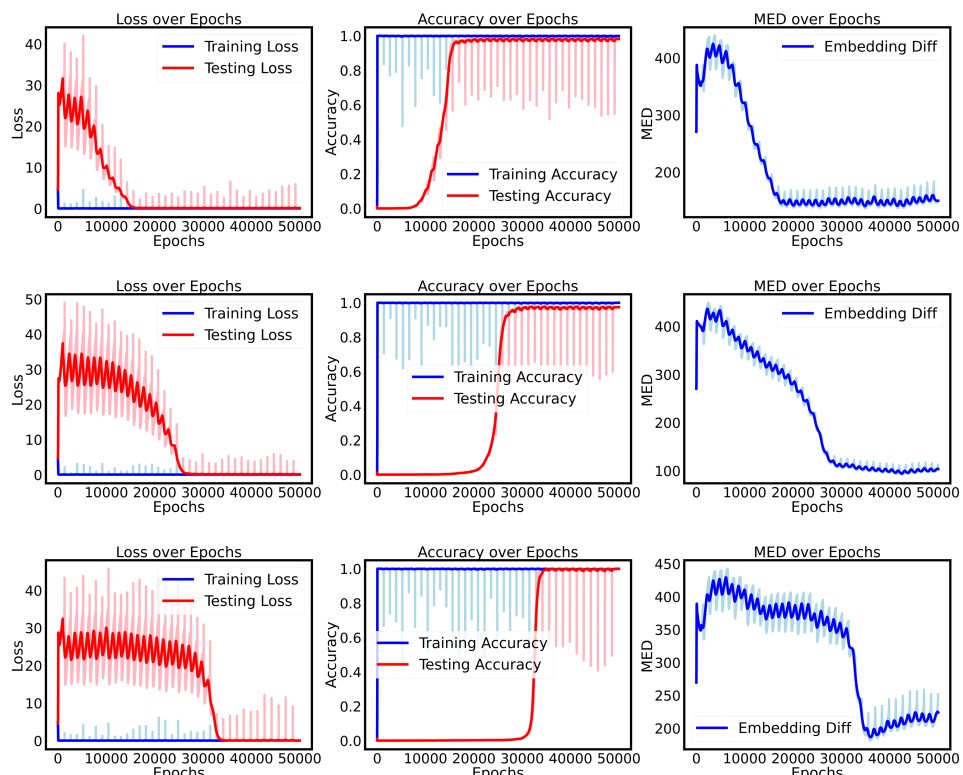

Figure 20: We arranged the images in order of generalization speed. However, in fact, positional encoding was not used in the first and third images, while it was used in the second image. This clearly demonstrates the instability of the generalization speed that we have discussed.

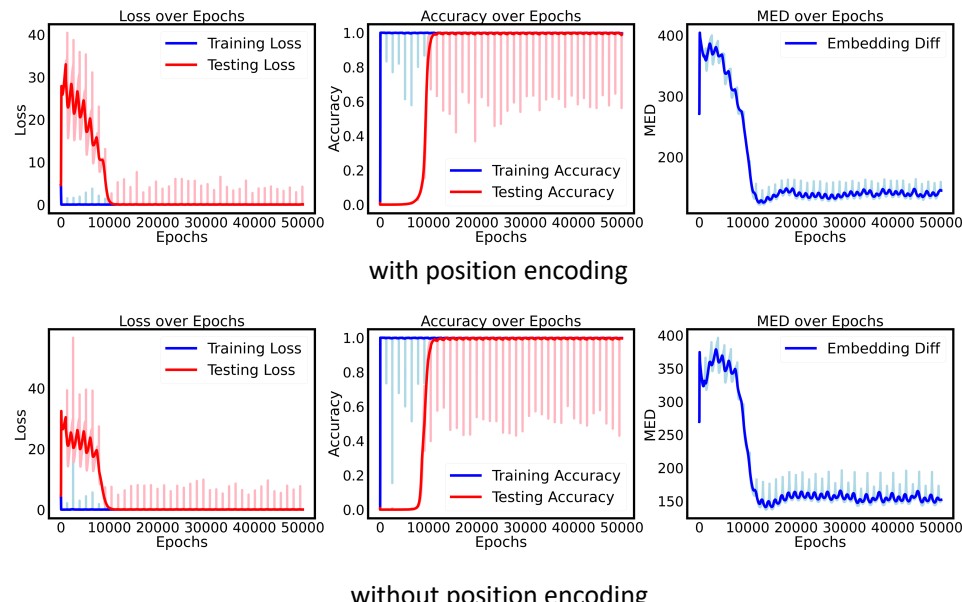

Figure 21: When the embedding dimension reaches 512, the instability disappears.

### E.2 DIFFERENCE IN THE UPPER BOUND OF GENERALIZATION UNDER BOUNDARY CONDITIONS

Another interesting phenomenon is that when the dataset size is in a boundary condition, smaller embedding dimensions may lead to different upper limits of achievable final test accuracy. We illustrate this scenario in Figure 22.

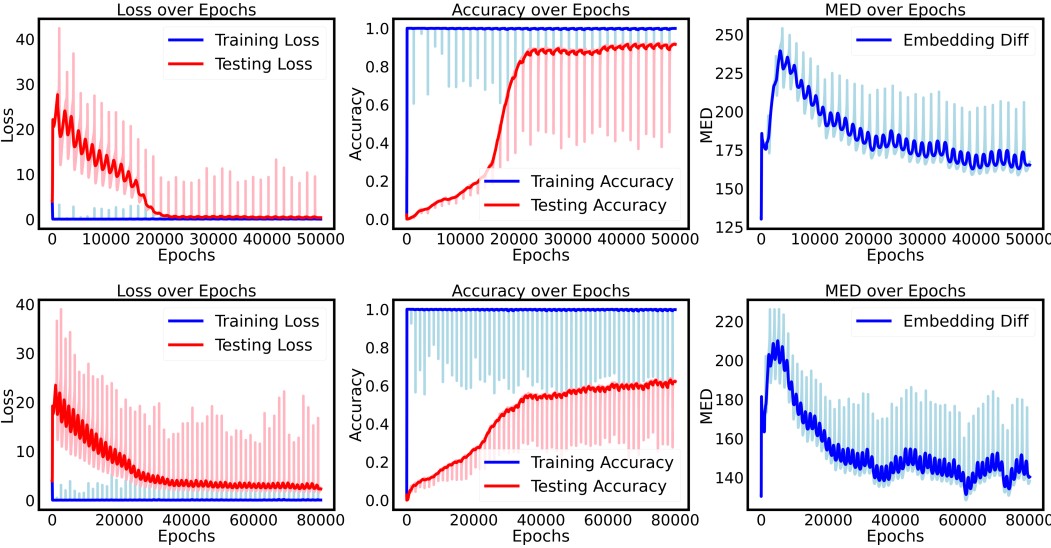

Figure 22: We selected $p = 47$, $k = 0$, $d_{model} = 128$ and the training set proportion is set to 0.3. Then we could see that between the two experiments, there are not only significant differences in the rate of convergence but also substantial discrepancies in the upper bound of achievable accuracy.

## F DISCUSSION ON THE INPUT REGION DISTRIBUTION

For conventional discussions on the normalization of embedding dictionaries, refer to Liu et al. (2022). Nanda et al. (2022) extends this conclusion to the overall structure. The foundation of

these theories is that we can interpret arithmetic operations over a prime field as rotations on a circle. Therefore, we might speculate whether the endpoint of the model's learning corresponds to an embedding dictionary aligned with such a circular structure. Our answer to this question is negative.

We consider the task $(a_i, b_i, p_+)$, $a_i \in \mathbb{A}$, $b_i \in \mathbb{B}$ with $\mathbb{A} \cup \mathbb{B} = \emptyset$, we calculate the diameter and the average neighbor distance of the embedding vector sets corresponding to $\mathbb{A}$ and $\mathbb{B}$ in this task.

We use $d$ to denote the diameter function. The experiments indicate that $d(\mathbb{A}, \mathbb{B}) \gg d(\mathbb{A}) + d(\mathbb{B})$. Moreover, the distances between the vast majority of neighboring embedding vectors are approximately half of the diameter, indicating that the embedding vectors cannot maintain a uniformly distributed ordered structure. Therefore, the distribution of our input region more closely resembles a covering space in topology rather than genuinely learning the structural characteristics of a prime field.

