# OpenReview forum: "Reconstruct the Understanding of Grokking through Dynamical Systems"
_ICLR.cc/2025/Conference — ICLR 2025 Conference Withdrawn Submission_

### Official Review · Reviewer_xfH7 · 2024-10-29

**Soundness:** 1
**Presentation:** 1
**Contribution:** 2
**Rating:** 1
**Confidence:** 3

**Summary:**

This work aims to ground the phenomenon of grokking in the mathematical framework of dynamical systems theory. The authors use the dynamical systems framing to develop a new progress measure that is independent of which prime is used for modular arithmetic. They use this insight to reduce grokking by choosing a specific embedding of the data.

**Strengths:**

The study of grokking is very relevant and the idea of using dynamical systems theory to study it is interesting. The reduction of the grokking (Figure 7) is promising, and the new progress measure could be of interest to the field.

**Weaknesses:**

There are several major weaknesses of this paper:

1. It was unclear to me what the embedding described in Section 3 is. Because this was unclear, much of the rest of the work was unclear. I would strongly recommend spending time making this section as clear as possible. In particular,
     a. Figure 2 says that $p$ symbols are used to represent the range $0$ to $p - 1$. What does this mean? Are these the embeddings, $\textbf{p}$. Is $p$ the same as the prime $p$ used in modular arthimetic?
      b. What is $c$ and $\gamma$? From context, I would assume that $c$ is the target output and $\gamma$ is the error, but this was never defined. What size is $c$?
      c. When is logit used?

2. The establishment of the model was also confusing. In particular, explaining what the sums in Eqs. 4-5 are over (and why there are sums) would be helpful in understanding what the authors did.

3. The authors make several assumptions that are not justified or motivated. For instance, Assumption 1 and 2 come out of nowhere. Similarly, saying $\alphda_d$ is "guessed to be approximately a linear function in specific situations" is very vague.

4. The main result (Theorem 4) is not stated mathematically and is vague. It was not clear what was really being gained from the theorem, nor was it clear how this was proven.

5. The progress measure (Eq. 12) is defined to be a function of $n$ but $n$ is not present in any of the equations. This makes it unclear what Pro 1 is saying.

6. "According to our theory, another factor that influences the rate of grokking is the size of d_model". It is not clear to me where the size of d_model comes in to the results presented in the main text.

7. The authors claim that increasing the size of the dataset affects the rate of grokking. But then they choose to increase $p$ and not $\beta$. The rationale for this was not clear. Additionally, given the bounds given in Sec, 5.2.2. $\beta$ and $p$ are in the numerator. Therefore, increasing either leads to greater norm, which I thought meant longer time to achive high test accuracy.

8. The mitigation of grokking (Figure 7) is interesting, but it is not clear how the authors use their previous results to choose a Toeplitz matrix. If anything, it feels like cheating slightly since the task is periodic and they choose a circulant matrix (or maybe not cheating, but rather they use features of the task instead of the dynamical systems picture they have developed).

9. There are a number of major typos, including sentences that do not make grammatical sense, sentences that end abruptly, and mis-spelling that makes reading this manuscript challenging.

**Questions:**

All questions are provided in the Weakness section.

---

### Official Review · Reviewer_zdeA · 2024-11-03

**Soundness:** 2
**Presentation:** 2
**Contribution:** 2
**Rating:** 5
**Confidence:** 3

**Summary:**

The authors propose a dynamical-systems view-point as a formal framework within which grokking can be understood. After discussing the use of general dynamical systems, specifically ODEs, in optimization, the manuscript proposes the study of a simple 3-D system as a reduced model capable of “identifying” grokking behavior. Several computational experiments are then discussed.

**Strengths:**

The proposed idea of studying grokking as emergent behavior from a dynamical system is in itself very interesting, and formal guarantees that could be drawn from such an analysis would greatly enhance our theoretical understanding of the phenomenon.

**Weaknesses:**

It is unclear how generalizable the analysis can be to arbitrary tasks, since it is applied to a very specific problem (even though the problem is, in some sense, canonical with respect to this phenomenon).

The manuscript offers a good formal description of the properties of the simple dynamical system. However, the connection between this projected system and its high-dimensional counterpart is not sufficiently motivated. This kind of discussion should be expanded.

Some minor concerns:
There are several small typos throughout the text.
Definition 2 is not clear and should be expanded
The statement “If the model have learnt” should be clarified. What does learning mean in a mathematical sense?

**Questions:**

Would one benefit from using a higher-dimensional “hidden” dynamical system that exhibits possibly more complicated behavior? Why are 3-dimensions always sufficient?

Why are the properties mentioned in the paragraph below the statement of Theorem 1 (line 233) desirable from a modelling perspective?

---

### Official Review · Reviewer_bxW2 · 2024-11-04

**Soundness:** 2
**Presentation:** 1
**Contribution:** 1
**Rating:** 1
**Confidence:** 4

**Summary:**

The paper presents an attempt an understanding and predicting 'Grokking", a curious phenomena in deep learning where test error rapidly decays long after training error appears to have converged. This paper models neural network parameter update as a dynamical system to 'reconstruct the analysis of grokking' and describes a metric which can 'predict' when grokking would occur during training.

**Strengths:**

Viewing gradient descent updates to neural network parameters as the evolution of a dynamical system is an interesting (but not novel) perspective. Trying to relate generalisation phenomena to properties of this system is a cool approach. The paper seems to be comprised of two parts. The first is a 'theoretical' component, where the authors make some deductions regarding the property of the dynamical system, and the second is a discussion of their "Main embedding diff", a measure of to track grokking.

**Weaknesses:**

I found this paper challenging to read. The language is unclear and the figures are repetitive and not particularly informative. While I understand the concept of modeling neural network updates as an ODE, it is not clear what is the exact dynamical system described in this paper. The lack of clarity is endemic to this work, as in all main equations in this paper (1-3, 4-6, and 7-9), it is left unexplained what their different components are. I believe that W & U are neural network parameters, but what are x, e, y, gamma, s, c, u, and w? The authors should make abundantly clear what are the roles of the different parameters in the system they model.
All this makes the "theoretical" part of this paper extremely hard to follow. While I could vaguely figure out the authors are attempting to make a claim regarding the existence of stable points relating to the neural network parameters, it is not obvious what that claim is, how they arrive at it, and why it is in any way relevant to grokking. The authors need to do a better job at explaining these extremely critical details.

In the second part, the authors introduce "Main embedding diff" (MED), a way of tracking grokking by measuring discrepancy between embeddings within a model. There seems to be no relationship between the first section and this one, and the paper is unclear as to what MED is, as the components laid out in eq 12 are not explained to the reader. The results consist of repetitions of the same figure presenting the grokking phenomena along with MED values during training for different model and task settings. Aside from MED being vaguely correlated with the grokking phenomena, it is not explained how it precisely relates to it. Furthermore, the paper only presents results for a single task (modular addition), while the original grokking work discusses several types of problems.

Overall, while the paper attempts to make theoretical contributions to understanding grokking, its unclear presentation, disconnected sections and weak results make it difficult to assess its actual scientific merit or practical value.

**Questions:**

Why do the learning curves in figures 3-7 have those dramatic spikes? These do not appear in the original Grokking work.

---

### Official Review · Reviewer_UxAN · 2024-11-04

**Soundness:** 1
**Presentation:** 1
**Contribution:** 1
**Rating:** 1
**Confidence:** 5

**Summary:**

The paper aims to study "grokking," or a delayed generalization phenomenon, using ideas from dynamical systems. The authors define a "hidden progress measure" that correlates with grokking and then propose a method to accelerate grokking.

**Strengths:**

The topic of delayed generalization is interesting and suitable to the ICLR audience.

**Weaknesses:**

I found this manuscript very difficult to follow and understand. There are many typos and the ideas are not clearly expressed. To give just a single example, Theorem 4 seems to contain no falsifiable statement. This is not to say there are no valuable ideas in the work, but as it stands I believe the work is well below the standards of publication for ICLR.

**Questions:**

NA

---

### Note · Authors · 2025-05-06

**Comment:**

Author is withdrawing due to the paper not meeting the desired quality standard.

**Withdrawal Confirmation:**

I have read and agree with the venue's withdrawal policy on behalf of myself and my co-authors.

---

### Meta-Review · Area_Chair_dkqp · 2024-12-19

**Metareview:**

This paper seeks to explain "grokking" -- which the authors define as a sudden, rapid rise in neural network performance after a large period of overfitting. There are a few problems raised by the reviewers: (1) there is a unanimous consensus that the paper is extremely challenging to read, with underdefined terms and vague theorem statements, and (2) it's not clear what in this paper generalizes. I think this second point is particularly important: it's not obvious to me what settings the authors' claimed phenomenon even occurs in. Many neural networks train as typical models, with test performance increasing rapidly early on in training and then slowing down as training converges. While it's true that test error continues to increase with training even once training error has reached zero, it's not obvious that this phenomenon is markedly more rapid than prior training of the model.

**Additional Comments On Reviewer Discussion:**

There was no reviewer discussion, as the authors did not post author feedback.

---

### Decision · Program_Chairs · 2025-01-22

Reject